# RFold: RNA Secondary Structure Prediction with Decoupled Optimization

## Abstract

The secondary structure of ribonucleic acid (RNA) is more stable and accessible in the cell than its tertiary structure, making it essential for functional prediction. Although deep learning has shown promising results in this field, current methods suffer from poor generalization and high complexity. In this work, we present RFold, a simple yet effective RNA secondary structure prediction in an end-to-end manner. RFold introduces a decoupled optimization process that decomposes the vanilla constraint satisfaction problem into row-wise and column-wise optimization, simplifying the solving process while guaranteeing the validity of the output. Moreover, RFold adopts attention maps as informative representations instead of designing hand-crafted features. Extensive experiments demonstrate that RFold achieves competitive performance and about eight times faster inference efficiency than the state-of-the-art method.

## 1 Introduction

Ribonucleic acid is essential in structural biology for its diverse functional classes [8, 45, 49, 18]. The functions of RNA molecules are determined by their structure [57]. The secondary structure, which contains the nucleotide base pairing information, as shown in Fig. 1, is crucial for the correct functions of RNA molecules [13, 11, 68]. Although experimental assays such as X-ray crystallography [6], nuclear magnetic resonance (NMR) [15], and cryogenic electron microscopy [12] can be implemented to determine RNA secondary structure, they suffer from low throughput and expensive cost.

Computational RNA secondary structure prediction methods have become increasingly popular due to their high efficiency [31]. Currently, these methods can be broadly classified into two categories [50, 14, 55, 60]: (i) comparative sequence analysis and (ii) single sequence folding algorithm. Comparative sequence analysis determines the secondary structure conserved among homologous sequences but the limited known RNA families hinder its development [35, 36, 28, 22, 21, 16, 43]. Researchers thus resort to single RNA sequence folding algorithms that do not need multiple sequence alignment information. A classical category of computational RNA folding algorithms is to use dynamic programming (DP) that assumes the secondary structure is a result of energy minimization [3, 44, 39, 73, 42, 10]. However, energy-based

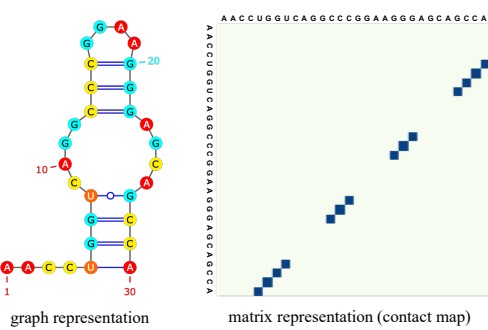

graph representation      matrix representation (contact map)

Figure 1: The graph and matrix representation of an RNA secondary structure example.

approaches usually require a nested structure, which ignores some biologically essential structures such as pseudoknots, i.e., non-nested base pairs [5, 54, 70], as shown in Fig. 2. Since predicting secondary structures with pseudoknots under the energy minimization framework has shown to be hard and NP-complete [66, 14], deep learning techniques are introduced as an alternative approach.

Attempts to overcome the limitations of energy-based methods have motivated deep learning methods in the absence of DP. SPOT-RNA [55] is a seminal work that ensembles ResNet [24] and LSTM [25] to identify molecular features. SPOT-RNA does not constrain the output space into valid RNA secondary structures, which degrades its generalization ability [32]. E2Efold [5] employs an unrolled algorithm for constrained programming that post-processes the network output to satisfy the constraints. E2Efold introduces a convex relaxation to make the optimization tractable, leading to possible constraint violations and poor generalization ability [53, 14]. Developing an appropriate optimization that forces the output to be valid becomes an important issue. Apart from the optimization problem, state-of-the-art approaches require hand-crafted features and introduce the pre-processing step for such features, which is inefficient and needs expert knowledge. CDPfold [72] develops a matrix representation based on sequence pairing that reflects the implicit matching between bases. UFold [14] follows the exact post-process mechanism as E2Efold and uses hand-crafted features from CDPfold with U-Net [51] model architecture to improve the performance.

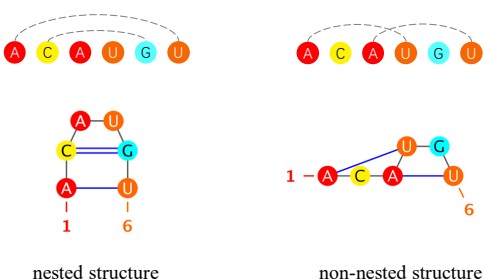

Figure 2: Examples of nested and non-nested secondary structures.

Although promising, current deep learning methods on RNA secondary structure prediction have been distressed by: (1) *the optimization process that is complicated and poor in generalization* and (2) *the data pre-processing that requires expensive complexity and expert knowledge*. In this paper, we present RFold, a simple yet effective RNA secondary structure prediction method in an end-to-end manner. Specifically, we introduce a decoupled optimization process that decomposes the vanilla constraint satisfaction problem into row-wise and column-wise optimization, simplifying the solving process while guaranteeing the validity of the output. Besides, we adopt attention maps as informative representations to automatically learn the pair-wise interactions of the nucleotide bases instead of using hand-crafted features to perform data pre-processing. We conduct extensive experiments to compare RFold with state-of-the-art methods on several benchmark datasets and show the superior performance of our proposed method. Moreover, RFold has faster inference efficiency than those methods due to its simplicity.

## 2 Related work

### 2.1 Comparative Sequence Analysis

Comparative sequence analysis determines base pairs conserved among homologous sequences [55, 17, 28, 36, 35, 19, 20]. ILM [52] combines thermodynamic and mutual information content scores. Sankoff [27] merges the sequence alignment and maximal-pairing folding methods [46]. Dynalign [41] and Carnac [62, 47] are the subsequent variants of Sankoff algorithms. RNA forester [26] introduces a tree alignment model for global and local alignments. However, the limited number of known RNA families [21, 16, 43] impedes the development of comparative methods.

### 2.2 Energy-based Folding Algorithms

When the secondary structure consists only of nested base pairing, dynamic programming can efficiently predict the structure by minimizing energy. Early works in this category include Vienna RNAfold [39], Mfold [73], RNAstructure [42], and CONTRAfold [10]. Faster implementations that speed up dynamic programming have been proposed, such as Vienna RNAplfold [4], LocalFold [37], and LinearFold [30]. However, these methods cannot accurately predict secondary structures with pseudoknots, as predicting the lowest free energy structures with pseudoknots is NP-complete [40], making it difficult to improve performance.

## 2.3 Learning-based Folding Algorithms

SPOT-RNA [55] is a seminal work that employs deep learning for RNA secondary structure prediction. SPOT-RNA2 [56] improves its predecessor by using evolution-derived sequence profiles and mutational coupling. Inspired by Raptor-X [65] and SPOT-Contact [23], SPOT-RNA uses ResNet and bidirectional LSTM with a sigmoid function to output the secondary structures. MXfold [1] is also an early work that combines support vector machines and thermodynamic models. CDPfold [72], DMFold [64], and MXFold2 [53] integrate deep learning techniques with energy-based methods. E2Efold [5] takes a remarkable step in constraining the output to be valid by learning unrolled algorithms. However, its relaxation for making the optimization tractable may violate the structural constraints. UFold [14] further introduces U-Net model architecture to improve performance.

# 3 Preliminaries and Backgrounds

## 3.1 Preliminaries

The primary structure of RNA is the ordered linear sequence of bases, which is typically represented as a string of letters. Formally, an RNA sequence can be represented as $\boldsymbol{X} = (x_1, ..., x_L)$, where $x_i \in \{A, U, C, G\}$ denotes one of the four bases, i.e., *Adenine* (A), *Uracil* (U), *Cytosine* (C), and *Guanine* (G). The secondary structure of RNA is a contact map represented as a matrix $\boldsymbol{M} \in \{0, 1\}^{L \times L}$, where $\boldsymbol{M}_{ij} = 1$ if the $i$-th and $j$-th bases are paired. In the RNA secondary structure prediction problem, we aim to obtain a model with learnable parameters $\Theta$ that learns a mapping $\mathcal{F}_\Theta : \boldsymbol{X} \mapsto \boldsymbol{M}$ by exploring the interactions between bases. Here, we decompose the mapping $\mathcal{F}_\Theta$ into two sub-mappings as:

$$\mathcal{F}_\Theta := \mathcal{H}_{\theta_h} \circ \mathcal{G}_{\theta_g}, \tag{1}$$

where $\mathcal{H}_{\theta_h} : \boldsymbol{X} \mapsto \boldsymbol{H}$, $\mathcal{G}_{\theta_g} : \boldsymbol{H} \mapsto \boldsymbol{M}$ are mappings parameterized by $\theta_h$ and $\theta_g$, respectively. $\boldsymbol{H} \in \mathbb{R}^{L \times L}$ is regarded as the unconstrained output of neural networks.

## 3.2 Backgrounds

It is worth noting that there are hard constraints on the formation of RNA secondary structure, meaning that certain types of pairing are not available [59]. Such constraints [5] can be formally described as follows:

- (a) Only three types of nucleotide combinations can form base pairs: $\mathcal{B} := \{AU, UA\} \cup \{GC, CG\} \cup \{GU, UG\}$. For any base pair $x_i x_j$ where $x_i x_j \notin \mathcal{B}$, $\boldsymbol{M}_{ij} = 0$.
- (b) No sharp loops within three bases. For any adjacent bases, there can be no pairing between them, i.e., $\forall |i - j| \leqslant 3, \boldsymbol{M}_{ij} = 0$.
- (c) There can be at most one pair for each base, i.e., $\forall i, \sum_{j=1}^{L} \boldsymbol{M}_{ij} \leqslant 1$.

The available space of valid secondary structures is all *symmetric* matrices $\in \{0, 1\}^{L \times L}$ that satisfy the above three constraints. The first two constraints can be satisfied easily. We define a constraint matrix $\overline{\boldsymbol{M}}$ as: $\overline{\boldsymbol{M}}_{ij} := 1$ if $x_i x_j \in \mathcal{B}$ and $|i - j| \geqslant 4$, and $\overline{\boldsymbol{M}}_{ij} := 0$ otherwise. By element-wise multiplication of the network output and the constraint matrix $\overline{\boldsymbol{M}}$, invalid pairs are masked.

The critical issue in obtaining a valid RNA secondary structure is the third constraint, i.e., *processing the network output to create a symmetric binary matrix that only allows a single "1" to exist in each row and column.* There are different strategies for dealing with this issue.

**SPOT-RNA** is a typical kind of method that imposes minor constraints. It takes the original output of neural networks $\boldsymbol{H}$ and directly applies the $\mathrm{Sigmoid}$ function, assigning a value of 1 to those greater than 0.5 and 0 to those less than 0.5. This process can be represented as:

$$\mathcal{G}(\boldsymbol{H}) = \mathbb{1}_{[\mathrm{Sigmoid}(\boldsymbol{H}) > 0.5]} \odot \boldsymbol{H}. \tag{2}$$

Here, the offset term $s$ has been set to 0.5. No explicit constraints are imposed, and no additional parameters $\theta_g$ are required.

**E2Efold** formulates the problem with constrained optimization and introduces an intermediate variable $\widehat{M} \in \mathbb{R}^{L \times L}$. It aims to maximize the predefined score function:

$$\mathcal{S}(\widehat{M}, H) = \frac{1}{2}\left\langle H - s, \mathcal{T}(\widehat{M})\right\rangle - \rho\|\widehat{M}\|_1, \tag{3}$$

where $\mathcal{T}(\widehat{M}) = \frac{1}{2}(\widehat{M} \odot \widehat{M} + (\widehat{M} \odot \widehat{M})^T) \odot \overline{M}$ ensures the output is a symmetric matrix that satisfies the constraints (a-b), $s$ is an offset term that is set as $\log(9.0)$ here, $\langle \cdot, \cdot \rangle$ denotes matrix inner product and $\rho\|\widehat{M}\|_1$ is a $\ell_1$ penalty term to make the matrix to be sparse.

The constraint (c) is imposed by requiring Eq. 3 to satisfy $\mathcal{T}(\widehat{M})\mathbb{1} \leqslant \mathbb{1}$. Thus, Eq. 3 is rewritten as:

$$\mathcal{S}(\widehat{M}, H) = \min_{\boldsymbol{\lambda} \geqslant \mathbf{0}} \frac{1}{2}\left\langle H - s, \mathcal{T}(\widehat{M})\right\rangle - \rho\|\widehat{M}\|_1 \quad -\left\langle \boldsymbol{\lambda}, \mathrm{ReLU}(\mathcal{T}(\widehat{M})\mathbb{1} - \mathbb{1})\right\rangle, \tag{4}$$

where $\boldsymbol{\lambda} \in \mathbb{R}_+^L$ is a Lagrange multiplier.

Formally, this process can be represented as:

$$\mathcal{G}_{\theta_g}(H) = \mathcal{T}(\arg\max_{\widehat{M} \in \mathbb{R}^{L \times L}} \mathcal{S}(\widehat{M}, H)). \tag{5}$$

Though three constraints are explicitly imposed in E2Efold, this method requires iterative steps to approximate the valid solutions and cannot guarantee that the results are entirely valid. Moreover, it needs a set of parameters $\theta_g$ in this processing, making tuning the model complex.

# 4 RFold

## 4.1 Decoupled Optimization

We propose the following formulation for the constrained optimization problem in RNA secondary structure problem:

$$\min_{M} - \mathrm{tr}(M^T \widehat{H})$$

$$\text{s.t. } \sum_{j=1}^{L} M_{ij} \leqslant 1, \forall i; \ \sum_{i=1}^{L} M_{ij} \leqslant 1, \forall j, \tag{6}$$

where $\mathrm{tr}(M^T \widehat{H}) = \sum_{i=1}^{L}\sum_{j=1}^{L} M_{ij}\widehat{H}_{ij}$ represents the trace operation. The matrix $\widehat{H}$ is symmetrized based on the original network output $H$ while satisfying the constraints (a-b) in Sec. 3.2 by multiplying the constraint matrix $\overline{M}$, i.e., $\widehat{H} = (H \odot H^T) \odot \overline{M}$.

We then propose to decouple the optimization process into row-wise and column-wise optimizations, and define the corresponding selection schemes as $S_r$ and $S_c$ respectively:

$$S_r = \{S_r^1, S_r^2, ..., S_r^L\}, \ S_c = \{S_c^1, S_c^2, ..., S_c^L\}, \tag{7}$$

where $S_r^i \in \{0,1\}^L$ signifies the selection scheme on the $i$th row, and $S_c^j \in \{0,1\}^L$ represents the selection scheme on the $j$th column. The score function is defined as:

$$\mathcal{S}(S_r, S_c, \widehat{H}) = -\mathrm{tr}(M^T \widehat{H}), \tag{8}$$

where $S_r, S_c$ constitute the decomposition of $M$. The goal of the score function is to maximize the dot product of $M$ and $\widehat{H}$ in order to select the maximum value in $\widehat{H}$. Our proposed decoupled optimization reformulates the original constrained optimization problem in Equation 6 as follows:

$$\min_{S_r, S_c} \mathcal{S}(S_r, S_c)$$

$$\text{s.t. } \sum_{i=1}^{L} S_r^i \leqslant \mathbb{1}, \forall i; \ \sum_{j=1}^{L} S_c^j \leqslant \mathbb{1}, \forall j. \tag{9}$$

If the corresponding $\widehat{H}_{ij}$ have the highest score in its row $\{\widehat{H}_{ik}\}_{k=1}^L$ and its column $\{\widehat{H}_{kj}\}_{k=1}^L$, then $M_{ij} = 1$. By exploring the optimal $S_r$ and $S_c$, the chosen base pairs can be obtained by the optimal scheme $S = S_r \otimes S_c$.

### 4.2 Row-Col Argmax

With the proposed decoupled optimization, the optimal matrix can be easily obtained using the variant Argmax function:

$$\text{Row-Col-Argmax}(\widehat{\boldsymbol{H}}) = \text{Row-Argmax}(\widehat{\boldsymbol{H}}) \odot \text{Col-Argmax}(\widehat{\boldsymbol{H}}) \tag{10}$$

where Row-Argmax and Col-Argmax are row-wise and column-wise Argmax functions respectively:

$$\text{Row-Argmax}_{ij}(\widehat{\boldsymbol{H}}) = \begin{cases} 1, & \text{if } \max\{\widehat{\boldsymbol{H}}_{ik}\}_{k=1}^{L} = \widehat{\boldsymbol{H}}_{ij}, \\ 0, & \text{otherwise.} \end{cases}$$

$$\text{Col-Argmax}_{ij}(\widehat{\boldsymbol{H}}) = \begin{cases} 1, & \text{if } \max\{\widehat{\boldsymbol{H}}_{kj}\}_{k=1}^{L} = \widehat{\boldsymbol{H}}_{ij}, \\ 0, & \text{otherwise.} \end{cases} \tag{11}$$

**Theorem 1.** Given a symmetric matrix $\widehat{\boldsymbol{H}} \in \mathbb{R}^{L \times L}$, the matrix $\text{Row-Col-Argmax}(\widehat{\boldsymbol{H}})$ is also a symmetric matrix.

*Proof:* See Appendix C.1.

As shown in Fig. 3, taking a random symmetric $6 \times 6$ matrix as an example, we show the output matrics of Row-Argmax, Col-Argmax, and Row-Col-Argmax functions, respectively. The Row-Col Argmax selects the value that has the maximum value on both its row and column while keeping the output matrix symmetric.

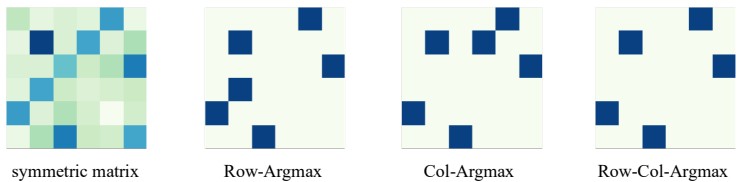

Figure 3: The visualization of the Row-Col-Argmax function.

From Theorem 1, we can observe that $\text{Row-Col-Argmax}(\widehat{\boldsymbol{H}})$ is a symmetric matrix that satisfies the constraint (c). Since $\widehat{\boldsymbol{H}}$ already satisfies constraints (a-b), the optimized output is:

$$\mathcal{G}(\boldsymbol{H}) = S_r \otimes S_c = \text{Row-Col-Argmax}(\widehat{\boldsymbol{H}}), \tag{12}$$

where $S_r, S_c = \arg\min_{S_r, S_c} -\text{tr}(S_r, S_c)$.

### 4.3 Row-Col Softmax

Though the Row-Col Argmax function can obtain the optimal matrix $\mathcal{G}(\boldsymbol{H})$, it is not differentiable and thus cannot be directly used in the training process. In the training phase, we need to use a differentiable function to approximate the optimal results. Therefore, we propose using a Row-Col Softmax function to approximate the Row-Col Argmax function for training. To achieve this, we perform row-wise Softmax and column-wise Softmax on the symmetric matrix $\widehat{\boldsymbol{H}}$ separately, as shown below:

$$\text{Row-Softmax}_{ij}(\widehat{\boldsymbol{H}}) = \frac{\exp(\widehat{\boldsymbol{H}}_{ij})}{\sum_{k=1}^{L} \exp(\widehat{\boldsymbol{H}}_{ik})},$$

$$\text{Col-Softmax}_{ij}(\widehat{\boldsymbol{H}}) = \frac{\exp(\widehat{\boldsymbol{H}}_{ij})}{\sum_{k=1}^{L} \exp(\widehat{\boldsymbol{H}}_{kj})}. \tag{13}$$

The Row-Col Softmax function is then defined as follows:

$$\text{Row-Col-Softmax}(\widehat{\boldsymbol{H}}) = \frac{1}{2}(\text{Row-Softmax}(\widehat{\boldsymbol{H}}) + \text{Col-Softmax}(\widehat{\boldsymbol{H}})), \tag{14}$$

Note that we use the average of $\text{Row-Softmax}(\widehat{\boldsymbol{H}})$ and $\text{Col-Softmax}(\widehat{\boldsymbol{H}})$ instead of the element product as shown in Equ. 10 for the convenience of optimization.

**Theorem 2.** Given a symmetric matrix $\widehat{\boldsymbol{H}} \in \mathbb{R}^{L \times L}$, the matrix Row-Col-Softmax$(\widehat{\boldsymbol{H}})$ is also a symmetric matrix.

*Proof:* See Appendix C.2.

As shown in Fig. 4, taking a random symmetric $6 \times 6$ matrix as an example, we show the output matrics of Row-Softmax, Col-Softmax, and Row-Col-Softmax functions, respectively. It can be seen that the output matrix of Row-Col-Softmax is still symmetric. Leveraging the differentiable property of Row-Col-Softmax, the model can be easily optimized.

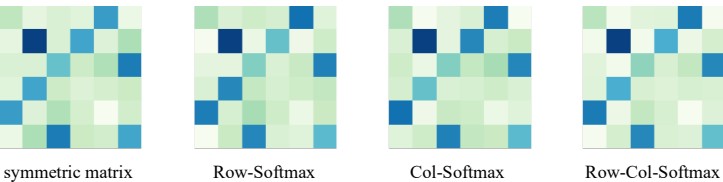

symmetric matrix     Row-Softmax     Col-Softmax     Row-Col-Softmax

Figure 4: The visualization of the Row-Col-Softmax function.

In the training phase, we apply the differentiable Row-Col Softmax activation and optimize the mean square error (MSE) loss function between $\mathcal{G}(\boldsymbol{H})$ and $\boldsymbol{M}$:

$$\mathcal{L}(\mathcal{G}(\boldsymbol{H}), \boldsymbol{M}) = \frac{1}{L^2} \|\text{Row-Col-Softmax}(\widehat{\boldsymbol{H}}) - \boldsymbol{M}\|^2. \tag{15}$$

### 4.4 Seq2map Attention

To simplify the pre-processing step that constructs hand-crafted features based on RNA sequences, we propose a Seq2map attention module that can automatically produce informative representations. We start with a sequence in the one-hot form $\boldsymbol{X} \in \mathbb{R}^{L \times 4}$ and obtain the sum of the token embedding and positional embedding as the input for the Seq2map attention. For convenience, we denote the input as $\boldsymbol{Z} \in \mathbb{R}^{L \times D}$, where $D$ is the hidden layer size of the token and positional embeddings.

Motivated by the recent progress in attention mechanisms [63, 9, 34, 7, 33, 48, 29, 69, 38], we aim to develop a highly effective sequence-to-map transformation based on pair-wise attention. We obtain the query $\boldsymbol{Q} \in \mathbb{R}^{L \times D}$ and key $\boldsymbol{K} \in \mathbb{R}^{L \times D}$ by applying per-dim scalars and offsets to $\boldsymbol{Z}$:

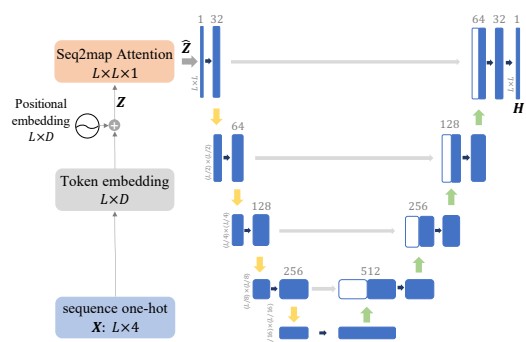

Figure 5: The overview model of RFold.

$$\boldsymbol{Q} = \gamma_Q \boldsymbol{Z} + \beta_Q, \quad \boldsymbol{K} = \gamma_K \boldsymbol{Z} + \beta_K, \tag{16}$$

where $\gamma_Q, \gamma_K, \beta_Q, \beta_K \in \mathbb{R}^{L \times D}$ are learnable parameters.

Then, the pair-wise attention map is obtained by:

$$\bar{\boldsymbol{Z}} = \text{ReLU}^2(\boldsymbol{Q}\boldsymbol{K}^T/L), \tag{17}$$

where $\text{ReLU}^2$ is an activation function that can be recognized as a simplified Softmax function in vanilla Transformers [58]. The output of Seq2map is the gated representation of $\bar{\boldsymbol{Z}}$:

$$\widehat{\boldsymbol{Z}} = \bar{\boldsymbol{Z}} \odot \sigma(\bar{\boldsymbol{Z}}), \tag{18}$$

where $\sigma(\cdot)$ is the Sigmoid function that performs as a gate operation.

As shown in Fig. 5, we identify the problem of predicting $\boldsymbol{H} \in \mathbb{R}^{L \times L}$ from the given sequence attention map $\widehat{\boldsymbol{Z}} \in \mathbb{R}^{L \times L}$ as an image-to-image segmentation problem and apply the U-Net model architecture to extract pair-wise information.

## 5 Experiments

We conduct experiments to compare our proposed RFold with state-of-the-art and commonly used methods in the field of RNA secondary structure prediction. Multiple experimental settings are taken into account, including standard RNA secondary structure prediction, generalization evaluation, large-scale benchmark evaluation, and inference time comparison. Ablation studies are also presented.

**Datasets** We use three benchmark datasets: (i) RNAStralign [61], one of the most comprehensive collections of RNA structures, is composed of 37,149 structures from 8 RNA types; (ii) ArchiveII [57], a widely used benchmark dataset in classical RNA folding methods, containing 3,975 RNA structures from 10 RNA types; (iii) bpRNA [55], is a large scale benchmark dataset, containing 102,318 structures from 2,588 RNA types.

**Baselines** We compare our proposed RFold with baselines including energy-based folding methods such as Mfold [73], RNAsoft [2], RNAfold [39], RNAstructure [42], CONTRAfold [10], Contextfold [71], and LinearFold [30]; learning-based folding methods such as SPOT-RNA [55], Externafold [67], E2Efold [5], MXfold2 [53], and UFold [14].

**Metrics** We evaluate the performance by precision, recall, and F1 score, which are defined as:

$$\text{Precision} = \frac{\text{TP}}{\text{TP} + \text{FP}}, \ \text{Recall} = \frac{\text{TP}}{\text{TP} + \text{FN}}, \ \text{F1} = 2\frac{\text{Precision} \cdot \text{Recall}}{\text{Precision} + \text{Recall}}, \tag{19}$$

where $\text{TP}, \text{FP}$, and $\text{FN}$ denote true positive, false positive and false negative, respectively.

**Implementation details** Following [14], we train the model for 100 epochs with the Adam optimizer. The learning rate is 0.001, and the batch size is 1 for sequences with different lengths.

### 5.1 Standard RNA Secondary Structure Prediction

Following [5], we split the RNAStralign dataset into training, validation, and testing sets by stratified sampling to ensure every set has all RNA types. We report the experimental results in Table 1. It can be seen that energy-based methods achieve relatively weak F1 scores ranging from 0.420 to 0.633. Learning-based folding algorithms like E2Efold and UFold can significantly improve performance by large margins, while RFold obtain even better performance among all the metrics. Moreover, RFold obtains about 8% higher precision than the state-of-the-art method. This phenomenon suggests that our proposed decoupled optimization is strict to satisfy all the hard constraints for predicting valid structures.

Table 1: Results on RNAStralign test set. Results in bold and underlined are the top-1 and top-2 performances, respectively.

| Method | Precision | Recall | F1 |
|---|---|---|---|
| Mfold | 0.450 | 0.398 | 0.420 |
| RNAfold | 0.516 | 0.568 | 0.540 |
| RNAstructure | 0.537 | 0.568 | 0.550 |
| CONTRAfold | 0.608 | 0.663 | 0.633 |
| LinearFold | 0.620 | 0.606 | 0.609 |
| CDPfold | 0.633 | 0.597 | 0.614 |
| E2Efold | 0.866 | 0.788 | 0.821 |
| UFold | 0.905 | 0.927 | 0.915 |
| RFold | **0.981** | **0.973** | **0.977** |

### 5.2 Generalization Evaluation

To verify the generalization ability of our proposed RFold, we directly evaluate the performance on another benchmark dataset ArchiveII using the pre-trained model on the RNAStralign training dataset. Following [5], we exclude RNA sequences in ArchiveII that have overlapping RNA types with the RNAStralign dataset for a fair comparison. The results are reported in Table 2.

It can be seen that traditional methods achieve F1 scores in the range of 0.545 to 0.842. Among the state-of-the-art methods, RFold attains the highest F1 score. It is noteworthy that RFold has a relatively lower recall metric and significantly higher precision metric. This phenomenon may be due to the strict constraints imposed by RFold. Although none of the current learning-based methods can meet all the constraints presented in Sec. 3.2, the predictions made by RFold are guaranteed to be valid. Therefore, RFold may cover fewer pairwise interactions, resulting in a lower recall metric. Nonetheless, the highest F1 score indicates the excellent generalization ability of RFold.

| Table 2: Results on ArchiveII dataset. | | | |
|---|---|---|---|
| Method | Precision | Recall | F1 |
| Mfold | 0.668 | 0.590 | 0.621 |
| RNAfold | 0.663 | 0.613 | 0.631 |
| RNAstructure | 0.664 | 0.606 | 0.628 |
| CONTRAfold | 0.696 | 0.651 | 0.665 |
| LinearFold | 0.724 | 0.605 | 0.647 |
| RNAsoft | 0.665 | 0.594 | 0.622 |
| Eternafold | 0.667 | 0.622 | 0.636 |
| E2Efold | 0.734 | 0.660 | 0.686 |
| SPOT-RNA | 0.743 | 0.726 | 0.711 |
| MXfold2 | 0.788 | 0.760 | 0.768 |
| Contextfold | 0.873 | 0.821 | 0.842 |
| UFold | 0.887 | **0.928** | 0.905 |
| RFold | **0.938** | 0.910 | **0.921** |

| Table 3: Results on bpRNA-TS0 set. | | | |
|---|---|---|---|
| Method | Precision | Recall | F1 |
| Mfold | 0.501 | 0.627 | 0.538 |
| E2Efold | 0.140 | 0.129 | 0.130 |
| RNAstructure | 0.494 | 0.622 | 0.533 |
| RNAsoft | 0.497 | 0.626 | 0.535 |
| RNAfold | 0.494 | 0.631 | 0.536 |
| Contextfold | 0.529 | 0.607 | 0.546 |
| LinearFold | 0.561 | 0.581 | 0.550 |
| MXfold2 | 0.519 | 0.646 | 0.558 |
| Externafold | 0.516 | 0.666 | 0.563 |
| CONTRAfold | 0.528 | 0.655 | 0.567 |
| SPOT-RNA | 0.594 | **0.693** | 0.619 |
| UFold | 0.521 | 0.588 | 0.553 |
| RFold | **0.692** | 0.635 | **0.644** |

## 5.3 Large-scale Benchmark Evaluation

The large-scale benchmark dataset bpRNA has a fixed training set (TR0), evaluation set (VL0), and testing set (TS0). Following [55, 53, 14], we train the model in bpRNA-TR0 and evaluate the performance on bpRNA-TS0 by using the best model learned from bpRNA-VL0. We summarize the evaluation results in Table 3. It can be seen that RFold significantly improves the previous state-of-the-art method SPOT-RNA by 4.0% in the F1 score.

Following [14], we conduct an experiment on long-range interactions. The bpRNA-TS0 dataset contains more versatile RNA sequences of different lengths and various types, which can be a reliable evaluation. Given a sequence of length $L$, the long-range base pairing is defined as the paired and unpaired bases with intervals longer than $L/2$. As shown in Table 4, RFold performs unexpectedly well on these long-range base pairing predictions. We can also find that UFold performs better in long-range cases than the complete cases. The possible reason may come from the U-Net model architecture that learns multi-scale features. RFold significantly improves UFold in all the metrics by large margins, demonstrating its strong predictive ability.

Table 4: Results on long-range bpRNA-TS0 set.

| Method | Precision | Recall | F1 |
|---|---|---|---|
| Mfold | 0.315 | 0.450 | 0.356 |
| RNAfold | 0.304 | 0.448 | 0.350 |
| RNAstructure | 0.299 | 0.428 | 0.339 |
| CONTRAfold | 0.306 | 0.439 | 0.349 |
| LinearFold | 0.281 | 0.355 | 0.305 |
| RNAsoft | 0.310 | 0.448 | 0.353 |
| Externafold | 0.308 | 0.458 | 0.355 |
| SPOT-RNA | 0.361 | 0.492 | 0.403 |
| MXfold2 | 0.318 | 0.450 | 0.360 |
| Contextfold | 0.332 | 0.432 | 0.363 |
| UFold | 0.543 | 0.631 | 0.584 |
| RFold | **0.803** | **0.765** | **0.701** |

## 5.4 Inference Time Comparison

We compared the running time of various methods for predicting RNA secondary structures using the RNAStralign testing set with the same experimental setting as in [14]. The results are presented in Table 5, which shows the average inference time per sequence. The fastest energy-based method is LinearFold, which takes an average of about 0.43s for each sequence. The previous learning-based baseline, UFold, takes about 0.16s. RFold has the highest inference speed, costing only about 0.02s per sequence. In particular, RFold is about eight times faster than UFold and sixteen times faster than MXfold2. The fast inference time of RFold is due to its simple sequence-to-map transformation.

Table 5: Inference time on the RNAStralign.

| Method | Time |
|---|---|
| CDPfold (Tensorflow) | 300.11 s |
| RNAstructure (C) | 142.02 s |
| CONTRAfold (C++) | 30.58 s |
| Mfold (C) | 7.65 s |
| Eternafold (C++) | 6.42 s |
| RNAsoft (C++) | 4.58 s |
| RNAfold (C) | 0.55 s |
| LinearFold (C++) | 0.43 s |
| SPOT-RNA(Pytorch) | 77.80 s (GPU) |
| E2Efold (Pytorch) | 0.40 s (GPU) |
| MXfold2 (Pytorch) | 0.31 s (GPU) |
| UFold (Pytorch) | 0.16 s (GPU) |
| RFold (Pytorch) | **0.02 s** (GPU) |

## 5.5 Ablation Study

**Decoupled Optimization**   To validate the effectiveness of our proposed decoupled optimization, we conduct an experiment that replaces them with other strategies. The results are summarized in Table 6, where RFold-E and RFold-S denote our model with the strategies of E2Efold and SPOT-RNA, respectively. We ignore the recent UFold because it follows exactly the same strategy as E2Efold. We also report the validity which is a sample-level metric evaluating whether all the constraints are satisfied. Though RFold-E has comparable performance in the first three metrics with ours, many of its predicted structures are invalid. The strategy of SPOT-RNA has incorporated no constraint that results in its low validity. Moreover, its strategy seems to not fit our model well, which may be caused by the simplicity of our RFold model.

Table 6: Ablation study on optimzation strategies (RNAStralign testing set).

| Method | Precision | Recall | F1 | Validity |
|---|---|---|---|---|
| RFold | **0.981** | **0.973** | **0.977** | **100.00**% |
| RFold-E | 0.888 | 0.906 | 0.896 | 50.31% |
| RFold-S | 0.223 | 0.988 | 0.353 | 0.00% |

Table 7: Ablation study on pre-processing strategies (RNAStralign testing set).

| Method | Precision | Recall | F1 | Time |
|---|---|---|---|---|
| RFold | **0.981** | **0.973** | **0.977** | 0.0167 |
| RFold-U | 0.875 | 0.941 | 0.906 | 0.0507 |
| RFold-SS | 0.886 | 0.945 | 0.913 | **0.0158** |

**Seq2map Attention**   We also conduct an experiment to evaluate the proposed Seq2map attention. We replace the Seq2map attention with the hand-crafted features from UFold and the outer concatenation from SPOT-RNA, which are denoted as RFold-U and RFold-SS, respectively. In addition to performance metrics, we also report the average inference time for each RNA sequence to evaluate the model complexity. We summarize the result in Table 7. It can be seen that RFold-U takes much more inference time than our RFold and RFold-SS due to the heavy computational cost when loading and learning from hand-crafted features. Moreover, it is surprising to find that RFold-SS has a little better performance than RFold-U, with the least inference time for its simple outer concatenation operation. However, neither RFold-U nor RFold-SS can provide informative representations.

## 5.6 Visualization

We visualize two examples predicted by RFold and UFold in Fig. 6. The corresponding F1 scores are denoted at the bottom of each plot. The first secondary structures is a simple example of a nested structure. It can be seen that UFold may fail in such a case. The second secondary structures is much more difficult that contains over 300 bases of the non-nested structure. While UFold fails in such a complex case, RFold can predict the structure accurately. Due to the limited space, we provide more visualization comparisons in Appendix D.

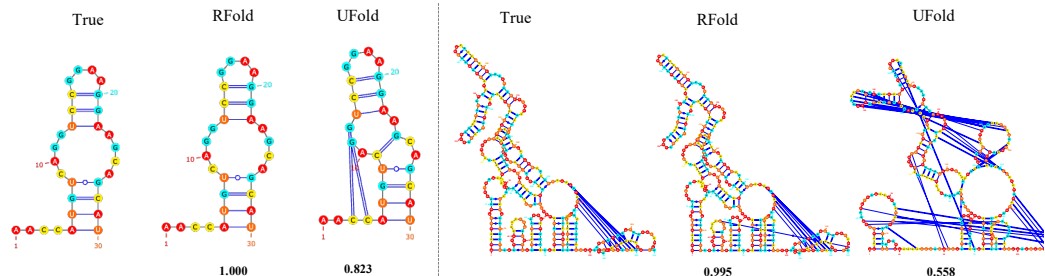

Figure 6: Visualization of the true and predicted structures.

## 6   Conclusion

In this study, we present RFold, a simple yet effective learning-based model for RNA secondary structure prediction. We propose decoupled optimization to replace the complicated post-processing strategies while incorporating constraints for the output. Seq2map attention is proposed for sequence-to-map transformation, which can automatically learn informative representations from a single sequence without extensive pre-processing operations. Comprehensive experiments demonstrate that RFold achieves competitive performance with faster inference speed. We hope RFold can provide a new perspective for efficient RNA secondary structure prediction.

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
