# OpenReview forum: "RFold: RNA Secondary Structure Prediction with Decoupled Optimization"
_NeurIPS.cc/2023/Conference — Submitted to NeurIPS 2023_

### Official Review · Reviewer_FGw2 · 2023-06-25

**Soundness:** 2 fair
**Presentation:** 3 good
**Contribution:** 3 good
**Rating:** 5
**Confidence:** 2

**Summary:**

In this paper, the authors propose a way to decouple the optimization process of RNA secondary structure prediction. Specifically, they decompose the constraint satisfaction problem into row-wise and column-wise optimization. Instead of hand-crafted features, attention maps are used to learn the pair-wise interactions of the nucleotide bases.

**Strengths:**

1. The authors show that it is more effective to use the attention maps as the input and then use U-Net to predict H, compared with using the hand-crafted features as input, which is interesting.
2. The proposed method reduces the inference time dramatically compared to various methods.
3. It achieves promising results on the RNAStralign dataset and large-scale benchmark evaluation.


**Weaknesses:**

1. The generalization ability is limited because the proposed method cannot achieve the best recall on ArchiveII and bpRNA-TS0 datasets.

**Questions:**

Could the authors use experiments to show that strict constraints cause the less promising generalization ability?
Is it possible to improve the proposed method to gain better generalization ability?

**Limitations:**

I did not find the potential negative societal impact of this work.

---

> ### Author Rebuttal · Authors · 2023-08-04
>
> Dear Reviewer FGw2,
>
> Thank you for your constructive and insightful comments! We appreciate the time and effort you've put into this review and would like to sincerely address your concerns below:
>
> ***
>
> **Q1** The proposed method cannot achieve the best recall on ArchiveII and bpRNA-TS0 datasets. Is it possible to improve the proposed method?
>
> **A1** The stringent constraints imposed by our model may result in a lower recall metric, suggesting that some uncertain pairs might be ignored due to these strict rules. In contrast, our precision metrics on ArchiveII and bpRNA-TS0 datasets are much higher, indicating that when RFold predicts a positive, it is likely to be accurate.
>
> Your question is constructive and valuable! The low recall of RFold may be due to the existence of abnormal samples. Although we have illustrated three hard constraints, some abnormal samples that do not satisfy these constraints do exist in practice. After analyzing the datasets used in this paper, we found that there are some abnormal samples in the test set. The ratio of valid samples in each dataset is summarized below:
>
> | Dataset/Method | Validity |
> |-------------|:---------:|
> | RNAStralign | 93.05% |
> | ArchiveII   | 96.03% |
> | bpRNA       | 96.51% |
>
> As shown in Table 6 of our manuscript, RFold enforces a validity of 100.00%, while other methods, like E2Efold, only achieve about 50.31%. RFold, therefore, more accurately reflects the real situation.
>
> We introduce a more flexible solution, soft-RFold, detailed in **Appendix B**. By incorporating a checking mechanism and adjusting the confidence threshold, soft-RFold is capable of enhancing the recall metric. Specifically, if the confidence given by the Rol-Col Softmax is low, we do not perform Rol-Col Argmax and assign more "1". The checking mechanism can be implemented as the following pseudo-code:
>
> ```
> y_pred = row_col_softmax(y)
> int_one = row_col_argmax(y_pred)
>
> # get the confidence for each position
> conf = y_pred * int_one
> all_pos = conf > 0.0
>
> # select reliable position
> conf_pos = conf > thr1
>
> # select unreliable position with the full row and column
> uncf_pos = get_unreliable_pos(all_pos, conf_pos)
>
> # assign "1" for the positions with the confidence higher than thr2
> # note that thr2 < thr1
> y_pred[uncf_pos] = (y_pred[uncf_pos] > thr2).float()
> int_one[uncf_pos] = y_pred[uncf_pos]
> ```
>
> We conduct experiments to compare the soft-RFold and the original RFold in the RNAStralign dataset:
>
> | Method | Precision | Recall | F1|
> |-------------|:---------:|:---------:|:---------:|
> | RFold | 0.981 | 0.973 | 0.977 |
> | soft-RFold | 0.978 | 0.974 | 0.976|
>
> It can be seen that soft-RFold improves the recall metric by a small margin. The minor improvement may be because the number of abnormal samples is small.
>
> We then select those samples that do not obey the three constraints to further analyse the performance. The total number of such samples is 179.
>
> | Method | Precision | Recall | F1|
> |-------------|:---------:|:---------:|:---------:|
> | RFold | 0.956 | 0.860 | 0.905 |
> | soft-RFold | 0.949 | 0.889 | 0.918|
>
> It can be seen that soft-RFold can deal with abnormal samples well. The improvement of the recall metric is more obvious.
>
> Our soft-RFold solution is quite intuitive, and we believe there is much room for exploration. We appreciate your valuable suggestion and will address this issue as part of our future work. Thanks a lot!
>
> ***

---

> > ### Comment · Reviewer_FGw2 · 2023-08-21
> >
> > I appreciate the clarification provided regarding the recall. It addressed my concerns.

---

### Official Review · Reviewer_G61p · 2023-07-03

**Soundness:** 2 fair
**Presentation:** 3 good
**Contribution:** 2 fair
**Rating:** 4
**Confidence:** 3

**Summary:**

The paper introduces RFold for RNA secondary structure prediction (a prediction of LxL binary matrix). It proposes to add a row-column-wise softmax at output of the model, before computing the L2 loss with respect to the ground truth. The experimental results show higher precision and recall compared to prior works.

**Strengths:**

- The paper is well-written with necessary backgrounds and basic introductions, problem formulations. Thus, this paper suited well for general audience of NeurIPS.
- RFold delivers strong performances in two commonly used datasets for evaluating RNA secondary structures.

**Weaknesses:**

The main concern is the limited novelty, RFold is incremental over Ufold. Both methods follow the paradigm of mapping a sequence RNA (using $\theta_{1}$) to $[L \times L \times n]$ features and further mapping the $[L\times L \times n]$ features (using $\theta_{2}$) to $[L \times L \times 1]$ output prediction. RFold differs from Ufold in two parts:
- RFold proposes $\theta_{1}$ to be represented by an attention-based layer. (RFold and Ufold use a similar, if not identical, $\theta_{2}$.)
- RFold adds a column-wise and a row-wised softmax after the $\theta_{2}$, before computing L2 loss.

Thus, RFold makes a few architectural modifications and improves the results.

E2Efold and Ufold also have a section on evaluation with pseudoknots on the RNAStralign test dataset, which this submission does not have.

**Questions:**

N/A

**Limitations:**

The paper has no section for limitations.

---

> ### Author Rebuttal · Authors · 2023-08-04
>
> Dear Reviewer G61p,
>
> Thank you for your thoughtful comments. We hope to address your concerns through the following responses.
>
>
> ***
>
> **Q1** Novelty may be limited.
>
> **A1** Thank you for your efforts in the review process. Please allow us to elaborate on our work.
>
> In general, we can divide deep-learning-based RNA secondary structure prediction methods into three key components: pre-processing, the backbone model, and post-processing:
> * The pre-processing step means projecting the 1D sequence into 2D matrix. (**1D -> discrete 2D**)
> * The backbone model learns from the 2D matrix and then output a hidden matrix of continuous values. (**discrete 2D -> continuous 2D**)
> * The post-processing step converts the hidden matrix into a contact map, which is matrix of discrete 0/1 values. (**continuous 2D -> discrete 2D**)
>
> To provide a clear methodology comparison, we have summarized the mainstream deep-learning-based methods based on the above key components. The definitions of constraints (a-c) used here are consistent with those in the paper:
>
> | Method | SPOT-RNA [1] | E2Efold [2] | UFold [3] | RFold |
> | --- | --- | --- | --- | --- |
> | Pre-Processing  | Pairwise concat | Pairwise concat | Pairwise concat + implicit matching [4] | Seq2map |
> | Backbone model  | CNN + LSTM | Transformer | U-Net | U-Net |
> | Post-Processing | Sigmoid($\cdot$)| Unrolled Algorithm | Unrolled Algorithm |  Decoupled Optimization |
> | Constraint (a)  | ✖️ | ✔️ | ✔️ | ✔️ |
> | Constraint (b)  | ✖️ | ✔️ | ✔️ | ✔️ |
> | Constraint (c)  | ✖️ | ✖️ | ✖️ | ✔️ |
>
> For the **pre-processing**, RFold is the unique approach that does not require explicit hand-crafted features produced by pairwise concat or implicit matching.
>
> For the **post-processing**, RFold propose a novel decoupled optimization to satisfy all constraints. While previous prevalent approaches such as E2EFold and UFold can only approximate, *they cannot guarantee satisfaction of constraint (c)*.
>
> Although RFold uses the same backbone model as UFold, the other two key components are significantly different from UFold. We admire the excellent work of UFold, which introduces more complex pre-processing and a stronger backbone model based on E2Efold to achieve strong performance. However, constraint (c) has not been addressed by UFold.
>
> Our work introduces decoupled optimization to satisfy all constraints, and proposes Seq2map to simplify the previously complex pre-processing, thereby achieving efficient RNA secondary structure prediction. *To the best of our knowledge, our proposed RFold is the first to solve the optimization problem that satisfies all constraints. Moreover, ours is the first approach to automate pre-processing rather than relying on manual design.*
>
> Additionally, as displayed in Table 6 of our manuscript, we noticed that the unrolled algorithm proposed by E2Efold, although it performs well on base-level metrics (precision, recall, and F1-score), only achieves about 50% on the sample-level validity metric. In contrast, our RFold method achieves high scores in both base-level and sample-level metrics.
>
> We believe that our proposed RFold is novel in RNA secondary structure prediction.
>
>
> ***
>
> **Q2** Evaluation with pseudoknots on the RNAStralign test dataset.
>
> **A2** Thank you for the thoughtful suggestion regarding pseudoknot evaluation. Following E2Efold and UFold, we counted the number of pseudoknotted sequences predicted as pseudoknotted, reporting this as true positives. We selected all pseudoknot-containing sequences from the RNAStralign test set. The results are as follows:
>
> | Method           | Precision | Recall | F1 score |
> | ---------------- | --------- | ------ | -------- |
> | RNAstructure [5] | 0.778     | 0.761  | 0.769    |
> | SPOT-RNA [1]     | 0.677     | 0.978  | 0.800    |
> | E2Efold [2]      | 0.844     | 0.990  | 0.911    |
> | UFold [3]        | 0.962     | 0.990  | 0.976    |
> | RFold            | **0.971** | **0.993** | **0.982** |
>
> Our proposed RFold exceeds previous state-of-the-art UFold across precision, recall, and F1 score, highlighting the effectiveness of our approach for modeling pseudoknotted structures.
>
> ***
>
> [1] Singh, Jaswinder, et al. RNA secondary structure prediction using an ensemble of two-dimensional deep neural networks and transfer learning. Nature communications, 2019.
>
> [2] Chen, Xinshi, et al. RNA secondary structure prediction by learning unrolled algorithms. ICLR, 2020.
>
> [3] Fu, Laiyi, et al. UFold: fast and accurate RNA secondary structure prediction with deep learning. Nucleic acids research, 2022.
>
> [4] Zhang, Hao, et al. A new method of RNA secondary structure prediction based on convolutional neural network and dynamic programming. Frontiers in genetics, 2019.
>
> [5] Mathews, David H., et al. Prediction of RNA secondary structure by free energy minimization. Current opinion in structural biology, 2006.

---

> > ### Comment · Reviewer_G61p · 2023-08-11
> >
> > I have read the rebuttal.

---

> > > ### Author Response · Authors · 2023-08-11
> > >
> > > Thank you for your insightful suggestions again.  We have included a more comprehensive comparison and experimental evidence as you suggested. We want to check in again and inquire if there are any other concerns we can address for you in order to potentially increase the evaluation score.

---

> > > ### Author Response · Authors · 2023-08-15
> > >
> > > Dear reviewer,
> > >
> > > We sincerely appreciate your valuable feedback and timely response.
> > >
> > > As the deadline for the author-reviewer discussion phase is approaching, we would like to check if you have any other remaining concerns about our paper. If our responses have adequately addressed your comments, we kindly request that you consider increasing the score.
> > >
> > > We sincerely thank you for your dedication and effort in evaluating our submission. Please do not hesitate to let us know if you need any clarification or have additional suggestions.
> > >
> > > Best Regards,
> > >
> > > Authors.

---

> > > > ### Comment · Reviewer_G61p · 2023-08-15
> > > >
> > > > Thank you for the authors' responses. I would like to begin by expressing my appreciation for the authors' efforts in clarifying the distinctions between the various methods. However, I remain somewhat unconvinced that the novelty of RFold holds substantial significance. The observed performance appears to be marginal at best, if not outright negligible, particularly on the RNAStralign test set (with only slight improvements observed in the ArchiveII dataset). The statistical significance of RFold's superiority over its counterparts remains unclear.
> > > >
> > > > As a result, I am inclined to maintain my score at 4, while adjusting my confidence level to 3.

---

> > > > > ### Author Response · Authors · 2023-08-15
> > > > >
> > > > >
> > > > > We sincerely appreciate your timely reply and are very pleased to engage in a detailed discussion with you.
> > > > >
> > > > > We respectfully **disagree with the assertion that our method's improvement is marginal**. Below, we have compiled a table summarizing the F1 scores and relative improvements achieved by both the current state-of-the-art method, UFold, and our method, RFold, across various datasets.
> > > > >
> > > > > | Method | RNAStralign | ArchiveII | bpRNA-TS0 | long-range bpRNA-TS0| Inference Time |
> > > > > |-|-|-|-|-|-|
> > > > > | UFold  | *0.915* | *0.905* | 0.553 | *0.584* | 0.16s|
> > > > > | RFold  | **0.977** | **0.921** | **0.644** | **0.701** | **0.02**s|
> > > > > | Relative Improvements| +6.78\% | +1.77\% | +16.46\% | +20.03\%| 8x|
> > > > >
> > > > > As can be seen from the above table, RFold has significant improvements across all datasets, with an average increase of 8.34%. Especially on RNAStralign and bpRNA-TS0, we achieved improvements of 6.78% and 16.46% respectively. On the long-range bpRNA-TS0, RFold even realized an impressive 20.03% improvement. Furthermore, RFold's inference time is one-eighth that of UFold. In summary, our method not only delivers substantial performance gains but also offers rapid inference speeds.
> > > > >
> > > > > It is important to underscore that **the F1-score has an upper bound of 1.00**. Considering that **gains become progressively harder to achieve as the F1-score exceeds 0.9**, we believe our results on RNAStralign are noteworthy. While the improvement on ArchiveII is modest at 1.77%, we attribute this to the dataset's smaller scale compared to the others, which renders its reference significance less robust.
> > > > >
> > > > > We greatly appreciate your timely response, and we sincerely hope that our explanation can sufficiently address your concerns.

---

> > > > > > ### Comment · Reviewer_G61p · 2023-08-21
> > > > > >
> > > > > > Thank you for clarification. My remaining concerns are (1) the work is incremental over Ufold and (2) the paper has no limitation section.

---

> > > > > > > ### Author Response · Authors · 2023-08-21
> > > > > > >
> > > > > > > Thanks for your reply!
> > > > > > >
> > > > > > > As today is the last day of the discussion phase, we would like to clarify the key issue as much as possible.
> > > > > > >
> > > > > > > As we mentioned during the rebuttal phase, **the only similarity between RFold and UFold is the U-Net backbone**. Both the pre-processing and post-processing strategies are completely different.
> > > > > > >
> > > > > > > In contrast, UFold employs the same post-processing strategy as E2EFold and a similar pre-processing approach. The significant difference from E2EFold is the utilization of U-Net. We highly respect this outstanding work, but in your opinion, it might be incremental work as well.
> > > > > > >
> > > > > > > We hope to clarify our work as clear as possible and deeply value your time and effort in the review process.

---

### Official Review · Reviewer_ZyLM · 2023-07-04

**Soundness:** 3 good
**Presentation:** 3 good
**Contribution:** 3 good
**Rating:** 7
**Confidence:** 4

**Summary:**

This work presents an efficient and accurate approach for end-to-end RNA secondary structure prediction.
The optimization problem formulation and its solution are well defined.
The results are strong and supported by visualizations and ablation studies.

**Strengths:**

The key strengths of this work are:
1. Inference is an order of magnitude faster than previous methods.
2. Inference is between 4-20% more accurate than previous methods, with significant gains specifically in long-range interactions.
3. A well defined optimization problem formulation and solution.
4. Including visualizations and ablation studies underscores the gains achieved through the optimization formulation and attention architecture.
5. The results are validated using multiple datasets and baselines.

**Weaknesses:**

Weaknesses of this work are:
1. There is a discrepancy in the definition of G in equation 12, where it does not incorporate the softmax function.
However, in equation 15, it is assumed as if it does. This can be fixed by introducing a new notation, such as G_{hat},
which includes the softmax function and will ensure consistency.
2. The definition of well-known metrics in section 5 is redundant.
3. The comparison between Rfold and Ufold could be more comprehensive,
describing their similarities and differences.
4. There are a few minor typos, and the writing may be improved.

**Questions:**

What are the key similarities and differences between Rfold and Ufold?
Visualizing performance as a function of sequence length may be useful.

**Limitations:**

The limitations are adequately addressed.

---

> ### Author Rebuttal · Authors · 2023-08-04
>
> Dear Reviewer RVaK,
>
> Thank you for your thoughtful and inspiring comment!
>
> ***
>
> **Q1** The key similarities and differences between Rfold and Ufold?
>
> **A1** Thank you for your meticulous review and insightful question! We can divide general deep-learning-based RNA secondary structure prediction methods into three key parts: pre-processing, the backbone model, and post-processing:
> * The pre-processing step means projecting the 1D sequence into 2D matrix. (**1D -> discrete 2D**)
> * The backbone model learns from the 2D matrix and then outputs a hidden matrix of continuous values. (**discrete 2D -> continuous 2D**)
> * The post-processing step converts the hidden matrix into a contact map, which is a matrix of discrete 0/1 values. (**continuous 2D -> discrete 2D**)
>
> Here, we will only introduce UFold and RFold. If you are interested in comparing other methods, please refer to the global response.
>
> We summarize the methodology comparison between UFold and RFold in the table below. The definitions of constraints (a-c) used here are consistent with those in the paper:
>
> | Method   | Pre-Processing | Backbone model | Post-Processing | Constraint (a) | Constraint (b) | Constraint (c) |
> | -------- | -------------- | -------------- | --------------- | -------------- | -------------- | -------------- |
> | UFold [1]| Pairwise concat + implicit matching [2] | U-Net | Unrolled Algorithm | ✔️ | ✔️ | ✖️ |
> | RFold    | Seq2map | U-Net | Decoupled Optimization | ✔️ | ✔️ | ✔️ |
>
> We delineate the key similarities and differences between RFold and UFold below:
>
> **Similarity**:
>
> *Backbone model*: Both RFold and UFold use the U-Net architecture as the backbone model.
>
> **Dissimilarity**:
>
> *Pre-processing*: UFold requires preprocessing of the RNA sequence into hand-crafted features, whereas RFold does not. Specifically, the input of UFold's U-Net is represented as a $17 \times L \times L$ matrix for a given RNA sequence of length $L$, while the input of RFold's U-Net is the feature map of $1 \times L \times L$ from the Seq2map Attention. RFold automatically obtains the feature map in a lightweight way.
>
> *Post-processing*: UFold employs the same post-processing strategy as E2Efold [3], which utilizes an unrolled algorithm. This approach, however, does not guarantee the satisfaction of the three important constraints. In contrast, RFold employs a decoupled optimization approach to satisfy these constraints in a simple and efficient manner.
>
> It can be seen that RFold only shares a similarity with UFold in the use of U-Net as the backbone model. The crucial aspects of pre-processing and post-processing differ significantly between the two.
>
> ***
>
> **Q2** Paper writing: (i) A discrepancy in the definition of G in equation 12; (ii) The definition of well-known metrics in Eq.(19) is redundant; (iii) A few minor typos.
>
> **A2**  We apologize for any confusion our writing may have caused, and we sincerely appreciate your detailed and valuable suggestions. We will thoroughly revise our paper based on your constructive comments.
>
> ***
>
>
> [1] Fu, Laiyi, et al. UFold: fast and accurate RNA secondary structure prediction with deep learning. Nucleic acids research, 2022.
>
> [2] Zhang, Hao, et al. A new method of RNA secondary structure prediction based on convolutional neural network and dynamic programming. Frontiers in genetics, 2019.
>
> [3] Chen, Xinshi, et al. RNA secondary structure prediction by learning unrolled algorithms. ICLR, 2020.

---

### Official Review · Reviewer_RVaK · 2023-07-25

**Soundness:** 2 fair
**Presentation:** 3 good
**Contribution:** 3 good
**Rating:** 5
**Confidence:** 3

**Summary:**

The paper proposes RFold, a simple and effective RNA secondary structure prediction algorithm. It adopts attention maps to learn informative representations for RNA rather than hand-crafted features. Then, based on a decoupled optimization process, RFold simplifies and guarantees satisfying the hard constraints on the formation of RNA secondary structure. Through the empirical experiments, the authors demonstrate that RFold achieves state-of-the-art performance with better computational efficiency compared to the previous works.

**Strengths:**

- The proposed decoupled optimization seems simple, but surprisingly effective for RNA secondary structure prediction. To the best of my knowledge, the proposed method is novel in the domain and might be promising for the broader machine-learning community.
- The proposed method shows great performance in three RNA benchmark datasets outperforming the previous state-of-the-art method by a significant margin. Some issues need to be addressed regarding the experiment setup (please refer to the weaknesses), but the improved performance seems truly impressive.

**Weaknesses:**

Major comments:
- [Data Split] To best approximate real-world applications that may require the prediction of novel structures, RNAs from the train/val/test set should bear minimal sequence and structural similarities. In contrast, it seems the authors have split datasets so that each RNA family has a similar fraction in each set. I think it may overestimate the true prediction performance of RFold. Likewise, “generalization to other datasets” experiments do not provide information about sequence/structure similarities between the datasets. If they are similar, it may not be a fair evaluation of generalization performance.
- Since the authors stated deep learning methods do not ignore the biologically essential structure such as pseudoknots, can you provide additional separate evaluation under the (non-) existence of pseudoknots?
- According to UFold, the bpRNA dataset contains mostly within family RNA species and does not adequately show the true generalization performance of the models. Can you provide additional evaluations with cross-family experiments?
- [Inference Time] It’s unclear whether the results are credible. The inference time can be quite different based on what type of machine (CPU, GPU, etc.) is used for the measurement. Since the other results seem to be excerpted from the UFold paper, the environments of UFold and RFold are likely to be different.
- [Reproduciblity] Architectural hyperparameters are missing. In addition, training codes do not seem to be included in the supplementary.

Minor comments:
- [Data Split] The authors stated that they split the RNAStralign dataset following the E2Efold paper. Can you confirm that all the methods including RFold used the same data splits? RNA sequences often have high sequence and structure similarities, so if you used different data splits it might affect the performance.
- As the authors stated, other algorithms often post-process the outputs to satisfy the constraints. Can you also show how the results are improved for RFold-E/S with the post-processing?

**Questions:**

- Do you have any plans for launching a web server for the proposed method? It would be difficult for many biotechnology researchers to set up the environment and run the algorithm. Therefore, while it is not mandatory, most compared algorithms support web servers for their models.

**Limitations:**

The authors have not discussed the limitations of the work.

---Post-Rebuttal Comments---

I appreciate the authors' dedication evident in their comprehensive responses. They have effectively addressed many of the concerns I had about the paper. Overall, while some concerns persist, I am inclined to believe that by incorporating the authors' responses, the manuscript's quality would be improved. Hence, I've adjusted my rating to 5.

---

> ### Author Rebuttal · Authors · 2023-08-04
>
> Dear Reviewer RVaK,
>
> Thanks for your professional and constructive comments! We respond to the questions as follows:
>
> ***
>
> **Q1** The data split might overestimate the prediction performance.
>
> **A1** We apologize for the confusion. We did not perform the data splitting ourselves. Rather, all the datasets used in this study come with their official splits, which we strictly adhered to in our experiments.
>
> In line 239, we stated, "*Following [5], we split the RNAStralign dataset into training, validation, and testing sets by stratified sampling.*" This was intended to clarify how the RNAStralign dataset was split. In practice, we directly use the data split from E2Efold. As far as we know, UFold used the same data split for the RNAStralign dataset.
>
> In summary, for both the RNAStralign and ArchiveII datasets, we followed the same data split as E2Efold and UFold, provided in the official E2Efold code. Regarding the bpRNA dataset, it comes with a predefined split (training on TR0, evaluating on VL0, testing on TS0), which we utilized directly in accordance with the official guidelines.
>
> We are sorry for the confusion and will refine the manuscript to make it clear.
>
> ***
>
>
> **Q2** Can you provide additional separate evaluations under the existence of pseudoknots?
>
> **A2** Thank you for your insightful and professional suggestion! Here, we present the evaluation of pseudoknot structure prediction. Following E2Efold and UFold, we count the number of pseudoknotted sequences that are predicted as pseudoknotted and report this count as true positive. We pick all sequences containing pseudoknots from the RNAStralign test dataset. The results are as follows:
>
> | Method           | Precision | Recall | F1 score |
> | ---------------- | --------- | ------ | -------- |
> | RNAstructure [4] | 0.778     | 0.761  | 0.769    |
> | SPOT-RNA [1]     | 0.677     | 0.978  | 0.800    |
> | E2Efold [2]      | 0.844     | 0.990  | 0.911    |
> | UFold [3]        | 0.962     | 0.990  | 0.976    |
> | RFold            | **0.971** | **0.993** | **0.982** |
>
> As the result demonstrates, RFold consistently surpasses UFold across all three metrics, indicating the effectiveness of our proposed approach.
>
> ***
>
>
> **Q3** Can you provide additional evaluations with cross-family experiments?
>
> **A3** Thank you for your professional insight! Initially, we did not include results from cross-family experiments as pure deep learning methods have struggled with this task. UFold, for instance, relies on the thermodynamic method Contrafold for data augmentation to achieve satisfactory results. Your valuable comment has made us realize the importance of including these results.
>
> We have conducted an evaluation using cross-family RNA from the bpRNA-new dataset. Notably, the standard UFold method achieves an F1 score of 0.583, while our RFold approach reaches 0.616. When the same data augmentation technique based on Contrafold [5] is applied, UFold's performance increases to 0.636, whereas our RFold method yields a score of 0.651. This places RFold second only to the thermodynamics-based method, Contrafold, in terms of F1 score.
>
> | Method | Precision | Recall | F1 score |
> | --- | --- | --- | --- |
> | E2Efold | 0.047 | 0.031 | 0.036 |
> | SPOT-RNA | 0.635 | 0.641 | 0.620 |
> | Contrafold | 0.620 | 0.736 | **0.661** |
> | UFold | 0.500 | 0.736 | 0.583 |
> | UFold + augmentation | 0.570 | 0.742 | 0.636|
> | RFold | 0.614 | 0.619 | 0.616 |
> | RFold + augmentation | 0.618 | 0.687 | ***0.651***|
>
> ***
>
> **Q4** Unclear whether the inference time is credible.
>
> **A4** We appreciate your thoughtful comment! For the comparison of inference time, we specifically rented an NVIDIA Titan Xp GPU to maintain consistency with the experimental setup of UFold. This information will be included in the revised manuscript.
>
> ***
>
>
> **Q5** Architectural hyperparameters are missing. The supplement provides the inference code but not the training code.
>
> **A5** We apologize for the lack of detailed architectural hyperparameters. For the seq2map attention, we employ a linear layer with a hidden size of 128. The U-Net backbone consists of four downsampling operations with max pooling in the encoding pathway, and four symmetric up-convolution blocks, each comprising an upsampling with a scale of 2 and a 2D convolution in the decoding pathway. The overall architecture aligns with that depicted in Figure 5 of the manuscript.
>
> The inference code includes the essential components of RFold. As our paper is still under review, we are not providing the training code at this time. However, we plan to make the training code available in the future.
>
> ***
>
>
> **Q6** Any plans for launching a web server?
>
> **A6** Thanks for your constructive suggestion!
>
> We should have open-sourced the inference code with pre-trained weights and Colab demo. All the experiments in this manuscript are reproduciable by simply running the code or the demo. However, according to the rules of NeurIPS, we are not allowed to share the links here. We will incorporate the establishment of a web server into our plan.
>
> ***
>
> [1] Singh, Jaswinder, et al. RNA secondary structure prediction using an ensemble of two-dimensional deep neural networks and transfer learning. Nature communications, 2019.
>
> [2] Chen, Xinshi, et al. RNA secondary structure prediction by learning unrolled algorithms. ICLR, 2020.
>
> [3] Fu, Laiyi, et al. UFold: fast and accurate RNA secondary structure prediction with deep learning. Nucleic acids research, 2022.
>
> [4] Mathews, David H., et al. Prediction of RNA secondary structure by free energy minimization. Current opinion in structural biology, 2006.
>
> [5] Do, Chuong B., et al. CONTRAfold: RNA secondary structure prediction without physics-based models. Bioinformatics, 2006.

---

> > ### Comment · Reviewer_RVaK · 2023-08-12
> > **Post-Rebuttal Comments**
> >
> > I appreciate the authors' dedication evident in their comprehensive responses. They have effectively addressed many of the concerns I had about the paper. Some of my remaining concerns are as follows:
> >
> > **Regarding Q3:** In my opinion, it's crucial to incorporate the results from the cross-family experiments and acknowledge the inherent limitations of the study. One aspect I'd like to highlight is that the current version of the paper might slightly overemphasize the generalization performance of the proposed method (as seen in Sec 5.2). It would greatly enhance the paper if the authors could revisit this section, discussing both the method's generalization capabilities and its limitations with a more cautious approach.
> >
> > **Regarding Q4:** I'm somewhat uncertain whether relying solely on the same GPU usage is sufficient to confidently assert the credibility of the inference time comparison. Given that numerous factors contribute to inference time and the time scale involved (< 1sec), results could be rather sensitive. Since the authors are exclusively focusing on inference time comparison, without factoring in training time, I believe it's imperative for them to measure the inference time across the compared methods in the same experiment setup. This will substantiate the claim regarding the efficiency of the proposed method.
> >
> > **Overall**, while some concerns persist, I am inclined to believe that by incorporating the authors' responses, the manuscript's quality would be improved. Hence, I've adjusted my rating to 5.

---

> > > ### Author Response · Authors · 2023-08-12
> > >
> > > Dear Reviewer,
> > >
> > > Thank you for your valuable feedback. We're pleased to note that several concerns have been addressed and truly appreciate your insightful and constructive comments!
> > >
> > > We will make detailed revisions based on your feedback in the final version.
> > >
> > > Best regards,
> > >
> > > Authors

---

### Author Rebuttal · Authors · 2023-08-04

We are grateful to the reviewers for their insightful and constructive feedback on our manuscript. We are encouraged by their recognition of our work as being **interesting and promising** (Reviewer RVaK, ZyLM, FGw2). Furthermore, the fact that they regard our methodology as **novel in the domain** (Reviewer RVaK) and **effective** (Reviewer RVaK, ZyLM, FGw2) is particularly encouraging. Additionally, the reviewers appreciated the **comprehensiveness of our experiments** (Reviewer RVaK, ZyLM, G61p, FGw2), and commented favorably on the **clarity of our presentation and the well-defined nature of our formulation** (Reviewer G61p, ZyLM).

In response to feedback, we provide detailed responses to address each reviewer’s concerns point by point. The response mainly includes:

### 1. Methodology Comparison

#### 1.1 General Comparison with Mainstream Methods

To provide a clear methodology comparison, we have summarized the mainstream methods alongside our own in the table below. The definitions of constraints (a-c) used here are consistent with those in the paper:

| Method | SPOT-RNA [1] | E2Efold [2] | UFold [3] | RFold |
| --- | --- | --- | --- | --- |
| Pre-Processing  | Pairwise concat | Pairwise concat | Pairwise concat + implicit matching [4] | Seq2map |
| Backbone model  | CNN + LSTM | Transformer | U-Net | U-Net |
| Post-Processing | Sigmoid($\cdot$)| Unrolled Algorithm | Unrolled Algorithm |  Decoupled Optimization |
| Constraint (a)  | ✖️ | ✔️ | ✔️ | ✔️ |
| Constraint (b)  | ✖️ | ✔️ | ✔️ | ✔️ |
| Constraint (c)  | ✖️ | ✖️ | ✖️ | ✔️ |

For the **pre-processing**, RFold is the unique approach that does not require explicit hand-crafted features produced by pairwise concat or implicit matching.

For the **post-processing**, RFold propose a novel decoupled optimization to satisfy all constraints. While previous prevalent approaches such as E2EFold and UFold can only approximate, *they cannot guarantee satisfaction of constraint (c)*.

#### 1.2 Detailed Comparison with UFold

As suggested by Reviewer ZyLM and G61p, we delineate the key similarities and differences between RFold and UFold below:

*Similarity*:

Both RFold and UFold use the U-Net architecture as the backbone model for RNA secondary structure prediction.

*Dissimilarity*:

(1) UFold requires preprocessing of the RNA sequence into hand-crafted features, whereas RFold does not. Specifically, the input of UFold's U-Net is represented as a $17 \times L \times L$ matrix for a given RNA sequence of length $L$, while the input of RFold's U-Net is the feature map of $1 \times L \times L$ from the Seq2map Attention.

(2) UFold employs the same post-processing strategy as E2Efold [3], which utilizes an unrolled algorithm. This approach, however, does not guarantee satisfaction of the three important constraints. In contrast, RFold employs a decoupled optimization approach to satisfy these constraints in a simple and efficient manner.

### 2. Evaluation with Pseudoknots

As recommended by Reviewers RVaK and G61p, we present the evaluation of pseudoknot structure prediction. We pick all sequences containing pseudoknots from the RNAStralign test dataset. The results are as follows:

| Method           | Precision | Recall | F1 score |
| ---------------- | --------- | ------ | -------- |
| HotKnots      | 0.500     | 0.565  | 0.531    |
| RNAstructure [5] | 0.778     | 0.761  | 0.769    |
| NuPack [6]       | 0.724     | 0.933  | 0.815    |
| SPOT-RNA [1]     | 0.677     | 0.978  | 0.800    |
| E2Efold [2]      | 0.844     | 0.990  | 0.911    |
| UFold [3]        | 0.962     | 0.990  | 0.976    |
| RFold            | **0.971** | **0.993** | **0.982** |

As can be seen, RFold consistently outperforms UFold across all three metrics, underscoring the efficacy of the proposed methodology.

### 3. Evaluation with Cross-family RNA

As suggested by Reviewer RVaK, we have conducted an evaluation using cross-family RNA from the bpRNA-new dataset. It's noteworthy that the standard UFold method attains an F1 score of 0.583, while our approach reaches 0.616. With the same data augmentation technique based on Contrafold [8], UFold's performance increases to 0.636, whereas our RFold method achieves 0.651. This F1 score positions RFold second only to the thermodynamics-based method, Contrafold.

| Method | Precision | Recall | F1 score |
| --- | --- | --- | --- |
| E2Efold | 0.047 | 0.031 | 0.036 |
| SPOT-RNA | 0.635 | 0.641 | 0.620 |
| Contrafold | 0.620 | 0.736 | **0.661** |
| UFold | 0.500 | 0.736 | 0.583 |
| UFold + augmentation | 0.570 | 0.742 | 0.636|
| RFold | 0.614 | 0.619 | 0.616 |
| RFold + augmentation | 0.618 | 0.687 | ***0.651***|

### 4. Dataset Split

In this work, we did not create any new datasets. Since the data and splits from previous studies are publicly available, we strictly adhered to their data and splits for our experiments.

### 5. Reproducibility

We should have open-sourced the inference code along with the pre-trained weights and provided a Colab demo for easy reproduction of all the experiments in this manuscript. However, in accordance with NeurIPS regulations, we are unable to share the links here.

We thank Reviewer RVaK for the great suggestion, and we will incorporate the establishment of a web server into our plan.

### 6. Constraints and Generalization Ability

The stringent constraints imposed by our model may result in a lower recall metric, suggesting that some uncertain pairs might be ignored due to these strict rules.

We introduce a more flexible solution, soft-RFold, detailed in Appendix B. By incorporating a checking mechanism and adjusting the confidence threshold, soft-RFold is capable of enhancing the recall metric.

---

### Author Response · Authors · 2023-08-15

Dear Reviewers,

We sincerely thank you for dedicating your valuable time to reviewing our manuscript. Your insightful comments and suggestions have been instrumental in refining the quality and clarity of our work.

We have thoroughly considered your thoughtful feedback and have carefully addressed each of your questions in our responses. We hope that our response has satisfactorily resolved your concerns. With the clarifications provided during the rebuttal process, we kindly hope you to raise the score for our paper, if you deem fit.

We look forward to your post-rebuttal feedback and sincerely thank you again for your contributions to strengthening our manuscript.

Best regards,

Authors

---

### Decision · Program_Chairs · 2023-09-21

**Decision:**

Reject

**Comment:**

This paper introduces a deep-learning based method for RNA secondary structure prediction with pseudoknots that enforces validity of the prediction (each base is paired to only one other base). Reviewers thought the method was interesting and had good performance, but had concerns about novelty and experimental results (in particular generalization performance, inference time evaluation and separate evaluation on pseudoknots). The authors provided comprehensive responses that addressed these issues to some extent, but issues regarding generalization performance of the proposed method and novelty remained after the discussion.

Overall, while the method is promising, the paper will require major revisions to address the issues highlighted by reviewers. In particular, the AC agrees that proper evaluation of the method in cross-family experiments and discussion of its limitations is critical to provide a realistic view of the method's performance. As such, the current version of the paper misses the bar for NeurIPS. That being said, the paper has potential and the AC encourages the authors to revise and resubmit it to a different venue.